# Trends in Health Care Access/Experiences: Differential Gains across Sexuality and Sex Intersections before and after Marriage Equality

**DOI:** 10.3390/ijerph19095075

**Published:** 2022-04-21

**Authors:** Rodman E. Turpin, Natasha D. Williams, Ellesse-Roselee L. Akré, Bradley O. Boekeloo, Jessica N. Fish

**Affiliations:** 1Department of Epidemiology and Biostatistics, School of Public Health, University of Maryland, College Park, MD 20742, USA; 2Department of Family Science, School of Public Health, University of Maryland, College Park, MD 20742, USA; will22@terpmail.umd.edu (N.D.W.); jnfish@umd.edu (J.N.F.); 3The Dartmouth Institute for Health Policy and Clinical Practice, Geisel School of Medicine, Dartmouth College, Hanover, NH 03755, USA; ellesse.l.akre@dartmouth.edu; 4Department of Behavioral and Community Health, School of Public Health, University of Maryland, College Park, MD 20742, USA; boekeloo@umd.edu

**Keywords:** LGBT, healthcare, national, barriers, services, time series

## Abstract

Background: Sexual minority adults experience several health care access inequities compared to their heterosexual peers; such inequities may be affected by LGBTQ+ legislation, such as the 2015 national marriage equality ruling. Methods: Using population-based data (*n* = 28,463) from the Association of American Medical Colleges biannual Consumer Survey of Health Care Access, we calculated trend ratios (TR) for indicators of health care access (e.g., insurance coverage, delaying or forgoing care due to cost) and satisfaction (e.g., general satisfaction, being mistreated due to sexual orientation) from 2013 to 2018 across sexuality and sex. We also tested for changes in trends related to the 2015 marriage equality ruling using interrupted time series trend interactions (TRInt). Results: The largest increases in access were observed in gay men (TR = 2.42, 95% CI 1.28, 4.57). Bisexual men had decreases in access over this period (TR = 0.47, 95% CI 0.22, 0.99). Only gay men had a significant increase in the health care access trend after U.S. national marriage equality (TRInt = 5.59, 95% CI 2.00, 9.18), while other sexual minority groups did not. Conclusions: We found that trends in health care access and satisfaction varied significantly across sexualities and sex. Our findings highlight important disparities in how federal marriage equality has benefited sexual minority groups.

## 1. Introduction

Sexual minority (i.e., lesbian, gay, and bisexual) adults experience a host of mental, behavioral, and physical health inequities compared to their heterosexual peers; these inequities have been linked to stigma and other social determinants of health [1,2]. Sexual minority adults are between 1.5 and 2.0 times more likely to report a mood or anxiety disorder and between 1.5 and 3.0 times more likely to meet the criteria for a substance use disorder when compared to their same-sex, heterosexual counterparts [3]. More recent studies highlight disparities in chronic health and all-cause mortality, including greater mortality for bisexual men and women relative to their heterosexual counterparts [4,5]. Sexual minority adults are at disproportionate risk for disordered weight [6], respiratory disorders [7], cardiovascular disease [8], and certain types of cancer [9]. Given these disparities, health care access, or a lack thereof, represents a critical barrier to health equity for sexual minority adults.

Research has consistently documented sexual-orientation-related disparities in health care access. Compared to their heterosexual peers, sexual minority adults are more likely to report unmet medical needs and less likely to receive routine medical care [10,11]. The disparity in health insurance coverage is a commonly cited factor in issues of health care access; sexual minority adults are less likely to report health insurance coverage compared to their heterosexual peers [12,13]. Cost is another common barrier to receiving health care, even among insured sexual minorities [14,15] Studies often find that sexual minorities delay or forgo care due to, in part, affordability [16,17]. Cost barriers persist despite insurance coverage gains that accompanied the passage of the Affordable Care Act and national marriage equality, though socioeconomic difficulties for sexual minorities vary substantially across regions and states, as well as across sexual identities [15,18]. 

Sexual minorities’ lack of satisfaction with health care services is a commonly studied barrier to care. Studies of overall satisfaction with medical care/providers consistently illustrate lower satisfaction levels among sexual minority patients relative to their heterosexual counterparts [15,19]. Sexual minority patients commonly report negative interactions with medical providers and staff when receiving care. These range from subtle, biased-based microaggressions to blatant heterosexism [20,21]. Given these experiences, sexual minority persons frequently experience discomfort when accessing health care and are reluctant to disclose their sexual identity [20,22]. Further, insufficient provider competence influences sexual minorities’ health-seeking behaviors, including how they search for providers and whether they seek care at all [16,23]. Even structural factors, such as the absence of state-level LGBTQ+ non-discrimination legislation, have been associated with lower patient satisfaction with their provider [24].

It is well-established that stigma at the individual (e.g., self-stigma), interpersonal (e.g., harassment, discrimination), and structural (e.g., discriminatory laws and policies) levels shape sexual minority health inequities [1]. Indeed, a compelling body of evidence now supports the link between stigma and mental health [3], substance use [25], and health status [26] among sexual minority populations across their life course [27]. Yet, sexual minority people have experienced swift changes in social acceptance and policy protections over the last decade [28]. Importantly, national health agencies and medical professions have become more aware of the unique health needs of this population. The implementation of the Affordable Care Act and the legalization of national marriage equality also led to important gains in health insurance coverage for sexual minority adults [29]. Thus, one would infer that sexual minority adults may have experienced substantial gains in health care access and reduced experiences of stigma when seeking health care. This is not entirely the case however, as despite the passage of national marriage equality, health care access inequities persist among many sexual minority groups. For this reason, it is important to further examine the role of national marriage equality in health care access and trends across sexual minority subgroups.

To address these research questions, we used a U.S. national population-based sample of health care consumers to assess trends from 2013 to 2018 in several indicators of health care access (e.g., insurance coverage, delaying or forgoing care due to cost) and satisfaction (e.g., general satisfaction, being mistreated due to sexual orientation) across groups defined by sexual identity and sex. We also assess the degree to which these changes may be related to the 2015 marriage equality ruling.

## 2. Materials and Methods

### 2.1. Sample

We conducted a secondary data analysis using data from the Association of American Medical Colleges (AAMC) biannual Consumer Survey of Health Care Access (CSHCA). This is a serial cross-sectional survey of a US national sample of respondents who reported needing health care in the last 12 months. This survey is conducted in two waves annually and each survey wave includes 2000–3500 respondents; Black, Hispanic, rural, and low-income populations and Medicaid recipients are oversampled in every other wave. Participants are recruited via e-mail and surveyed online. Survey weights were calculated based on sex, age, race and ethnicity, employment status, and household income to match the sociodemographics in the US adult population. All analyses used data from wave seven and later (11 waves from 2013 to 2018; *n* = 28,463) as these included sexual identity measures. Eligibility criteria included living in the United States, being 18 years of age or older, and indicating that either they or a healthcare professional believed they needed medical care in the past 12 months.

### 2.2. Sexual Identity, Sex, and Time

Sexual identity was measured using the question, “How do you self-identify? (heterosexual or straight, gay or lesbian, bisexual, other). Sex was measured using the question, “Are you male/female? (male, female). Based on these, we created an eight-category variable (heterosexual male, gay male, bisexual male, other male, heterosexual female, gay female, bisexual female, other female). For initial bivariate measures, we used a dichotomized measure of time, stratified in June 2015. The first five waves were included as pre-marriage equality (before June 2015), and the last six were included as post-marriage equality (June 2015 and after). For correlation and regression analyses, we used the uncategorized measure from the 11 waves, with each wave collected six months apart.

### 2.3. Health Care Access and Satisfaction

Access to services was assessed using six items: no insurance coverage over the past year; delaying getting medical care that the participant or a healthcare professional believed necessary; being unable to fill a prescription for medicine in the past year because of the out-of-pocket cost; skipping a medical test, treatment or follow-up recommended by a doctor because of the out-of-pocket cost; having problems paying for medical bills; and being unable to get needed medical care in the past year. Health care satisfaction was measured using three items: if the participant would recommend the provider who treated them during their most recent medical care visit to family and friends; if, during their visit, they were treated unfairly due to their sexual orientation; and if they were satisfied with the health care they received during their most recent medical care visit. From the aforementioned nine items, we created a single summed index. Each of the items was recoded such that 0 represented lower access/satisfaction, and 1 represented greater access/satisfaction. These were summed, creating an index ranging from 0 to 9, with higher scores representing increased access/satisfaction (i.e., fewer barriers to care). All items demonstrated good internal consistency (Cronbach’s alpha = 0.72). Additionally, factor analysis demonstrated a single factor for all items, consistent with unidimensionality.

### 2.4. Covariates

Covariates included the highest education level (less than high school, high school, college, graduate degree), employment status (full-time, part-time, retired, student/homemaker, unemployed), annual household income ($24,999 or less, $25,000 to $49,000, $50,000 to $74,999, $75,000 to $99,999, $100,000 or more), marital status (single, married, widowed, divorced, separated), and region (Northeast, Midwest, West, South).

### 2.5. Missing Data

Missingness was overall low for all items (<3%). We conducted intrascale stochastic imputation to impute missing observations for each health care access/satisfaction item from non-missing health care access/satisfaction items. This was appropriate given the low nonresponse for all variables and high internal consistency of the items. Similarly, missing socioeconomic covariates were imputed using all other socioeconomic covariates, as socioeconomic measures also demonstrated good internal consistency (Cronbach’s alpha = 0.74).

### 2.6. Analyses

For bivariate analyses, we tested if time post-marriage equality was associated with each health care/satisfaction measure using Chi-Square tests. For these analyses only, time was coded as a binary measure, with all time values grouped into one of two categories: Any time before June 2015, or June 2015 and after. We also tested for associations between the original discrete time measure and our health care access/satisfaction scale using a Spearman rank-sum correlation test. Finally, to visualize trends in health care access and satisfaction, we generated box plots with ordinal trendlines for our health care access/satisfaction scale. All bivariate analyses were stratified by sex and sexual identity to examine differences in health care access and satisfaction across sex and sexual identity. We also tested for trends in these health care outcomes using ordinal time series trend regression models. Cumulative ordinal trend ratios were generated, testing associations between time and our health care index, stratified by sex and sexual identity. To determine if health care access/satisfaction trends were significantly different across sex and sexual minority groups, we generated a single model with interaction terms, testing sex and sexual identity as an effect modifier of the trend in health care access. Trend ratios and 95% confidence intervals are reported. To examine if trends in health care access and satisfaction changed for sexual minority groups after marriage equality, we also used an interrupted time series regression model to test for a difference in the trends of health care access before and after the post-marriage equality cut-off point (June 2015). For all models, we included unadjusted estimates and estimates adjusted for education level, annual household income, employment status, marital status, and region. These covariates were included as they resulted in at least a 10% change in estimates, while other covariates (e.g., age, race/ethnicity) did not substantially change the findings (<5% change in the presented findings).

### 2.7. Quality Assurance and Statistical Software

We tested regression models for collinearity by measuring the variance inflation factor (VIF) in all models: There was no evidence of collinearity (all VIF < 5). We identified no influential outliers using Leverages and Cook’s distances. All analyses were conducted in 2021 using SAS 9.4 (SAS Institute Inc., Cary, NC, USA) [30].

## 3. Results

### 3.1. Sample Characteristics and Bivariate Results

Gay/lesbian, bisexual, and other sexual minority participants comprised 8.7% of our sample (Table 1). Comparing pre-national marriage equality to post-national marriage equality, gay men had significant increases in health care access across almost every single item and increased proportions of having no healthcare barriers (30.2% to 36.4%). The largest increase in proportions of having no healthcare barriers was observed among other sexual minority men (15.6% to 24.7%). In contrast, bisexual men had significant decreases in health care access across almost every single item and decreased proportions of having no healthcare barriers (26.9% to 19.4%). Gay/lesbian women had decreases in their ability to get prescriptions, medical tests, and pay for needed health care, as well as the largest decreases in fair treatment due to sexual orientation (88.9% to 68.3%). Heterosexual men and women had no significant changes in health care access between these two time periods.

### 3.2. Regression and Interactions

We identified differences in trends in health care access across sexual identity over the years 2013 to 2018 (Table 2). Overall, there was a significant interaction between trend time and combined gender and sexual identity measure. The largest increases in health care access over time were observed in gay men (TR = 2.42, 95% CI 1.28, 4.57), followed by bisexual women (TR = 1.45, 95% CI 1.02, 2.05), heterosexual men (TR = 1.21, 95% CI 1.08, 1.37), and heterosexual women (TR = 1.17, 95% CI 1.07, 1.29). These were largely unchanged after adjustment for socioeconomic factors, except for estimates for gay men (aTR = 1.88, 95% CI 1.06, 3.34). Bisexual men had decreases in health care access over time (TR = 0.47, 95% CI 0.22, 0.99), which was attenuated after adjustment for socioeconomic factors (aTR = 0.63, 95% CI 0.32, 1.22). Box-plot-based trend estimates were consistent with these findings (Figure 1). Additionally, education level, income, and proportions of married status all significantly increased over the time period (Appendix A). Notably, gay men had a significant increase in the health care access trend after U.S. national marriage equality, based on interrupted time series estimates (unadjusted = 5.59, 95% CI 2.00, 9.18; adjusted = 5.06, 95% CI 1.72, 8.40), while other sexual minority groups and heterosexuals did not (Table 3). Here, the interrupted time series estimates reflect the direction of the change in trends, and confidence intervals that exclude 1, indicate statistical significance. The positive estimate for gay men (5.06) indicates an increase in the trend in health care access, with statistical evidence (*p* < 0.05) that the slope trend in health care access became more positive after the marriage equality cut-off (June 2015).

## 4. Discussion

Our findings suggest that despite other research noting health care access improvements, many sexual minority groups in our sample have not experienced these health care gains, particularly post-marriage equality. Although there has been substantial social progress with regard to sexual minority visibility and rights, our findings indicate the greatest gains in health care access and satisfaction were limited to gay men, with gay men also having the only significant increase in health care access trends post-marriage equality. Sexual minority groups other than gay men may face unique barriers to health care access, such as the intersectional barriers to health care faced by sexual minority women [29]. Bisexual men and women often face greater health care disparities compared to their gay counterparts; this is evident in bisexual men being the only group with decreasing health care access over time. Marriage equality was a necessary but, alone, insufficient step to helping LGBTQ+ populations gain access to health insurance, but there needs to be clear and targeted policies and programs to address these additional barriers highlighted in our findings, such as cost and unfair treatment due to sexual orientation; this did not significantly improve for any sexual minority groups post-marriage equality. Additionally, marital status alone did not explain or account for the observed differences in health care access across the groups.

While we adjusted the measures for socioeconomic factors, it is important to note that this does not represent confounding, as the trend ratios reflect time as an exposure, so confounding is not a limitation here. Sexual identity is a moderator of the examined trend ratios. This does, however, allow us to explore if the observed trends are due to changes in socioeconomic factors over this time period. The attenuation of positive trends in health care access for gay men, after adjusting for socioeconomic factors, indicates that much of the health care access gains were due to socioeconomic improvements. Still, gay men maintained the largest positive trends in health care access over time, even after adjusting for these factors. Overall, socioeconomic factors do not completely explain the changes in health care access for any of the groups; there have been a number of studies that address how marriage equality has led to gains in insurance coverage, but less into how this policy ruling might have affected actual health-seeking behaviors, and other barriers to care that are not directly related to socioeconomic status [12,17]. In the context of developing policy to address health care access among sexual minority populations, our findings highlight both the relevance of socioeconomic factors, as well as the importance of addressing factors beyond just socioeconomics, such as health literacy, medical mistrust, and cultural competency of providers regarding LGBTQ+ populations [23]. Future research exploring the factors mediating the relationship between sexual minority identities and health care access, as well as subsequent adverse health outcomes, is recommended.

Our study had a number of strengths and limitations. First, this was a serial cross-sectional study; future research using prospective data is needed to assess within-person differences in health care access and satisfaction over time. Additionally, the data was limited to individuals who believed that they needed health care or were told by a physician that they were in need of health care; self-reported need is subjective and therefore may be subject to response bias. Moreover, the findings may not be generalizable to the general public and restricted only to those who need care. Despite this, our health care access outcomes are especially relevant to the population of those who need health care, more so than the general population. Another limitation with the data is that the years pooled for the analysis include time periods that are both before and after the full implementation of the Patient Protection and Affordable Care Act (ACA). While important provisions for accessing health care services were enacted in 2010, such as the removal of out-of-pocket costs for preventative services, other provisions that increased access to health insurance coverage and subsequently increased health care access, like Medicaid expansion in several states, did not take effect until 2014. Thus, respondents in waves prior to January 2014 are referencing experiences that occurred in a different health care landscape than respondents who are referencing health care experienced after the full implementation of the ACA. However, our study does highlight the benefits in trends in health care access post-marriage equality and is novel in highlighting inequities in health care access gains across different sexual minority groups. Our study was limited to sexual minority participants and did not capture the experiences of transgender people. The literature demonstrates that transgender people have different experiences in health care than cisgender people and it is essential that these differences are demonstrated in the literature; this is an important direction for future research. The focus on sexual minorities is a strength, however, as this is a population with well documented health inequities, making health care access an important factor to study. Minority stress, based on sexual minority identity, including discrimination and structural challenges, may not capture the full breadth of the stressors these minorities experience. Sexual minority individuals may also experience health care barriers related to race, or other identities. Future research into these intersectional identities and experiences with health care is recommended. Lastly, the study was limited in the ambiguity that the “other” sexual orientation category introduced. Given the small number of individuals in this group (particularly the “other men” group) it is difficult to make inferences about this population. The survey did not allow for respondents to clarify what they meant by “other.” Our focus on sexual minority subgroups is an important strength, however, as it shows how health care access varies substantially by sexual minority subgroup.

## 5. Conclusions

Overall, we found that trends in health care access and satisfaction from 2013 to 2018 varied significantly across sexual identities, with gay men, bisexual women, and heterosexual men and women having increasing health care access and satisfaction. Notably, gay men were the only group to have a significant increase in health care access trends post-marriage equality. Bisexual men were the only identity to have significantly decreasing trends in health care access. Our findings highlight important disparities in how federal marriage equality policy has benefited sexual minority groups. Further research into how marriage equality affects health-seeking behaviors and other cost-related barriers to care is recommended to help elucidate our findings.

## Figures and Tables

**Figure 1 ijerph-19-05075-f001:**
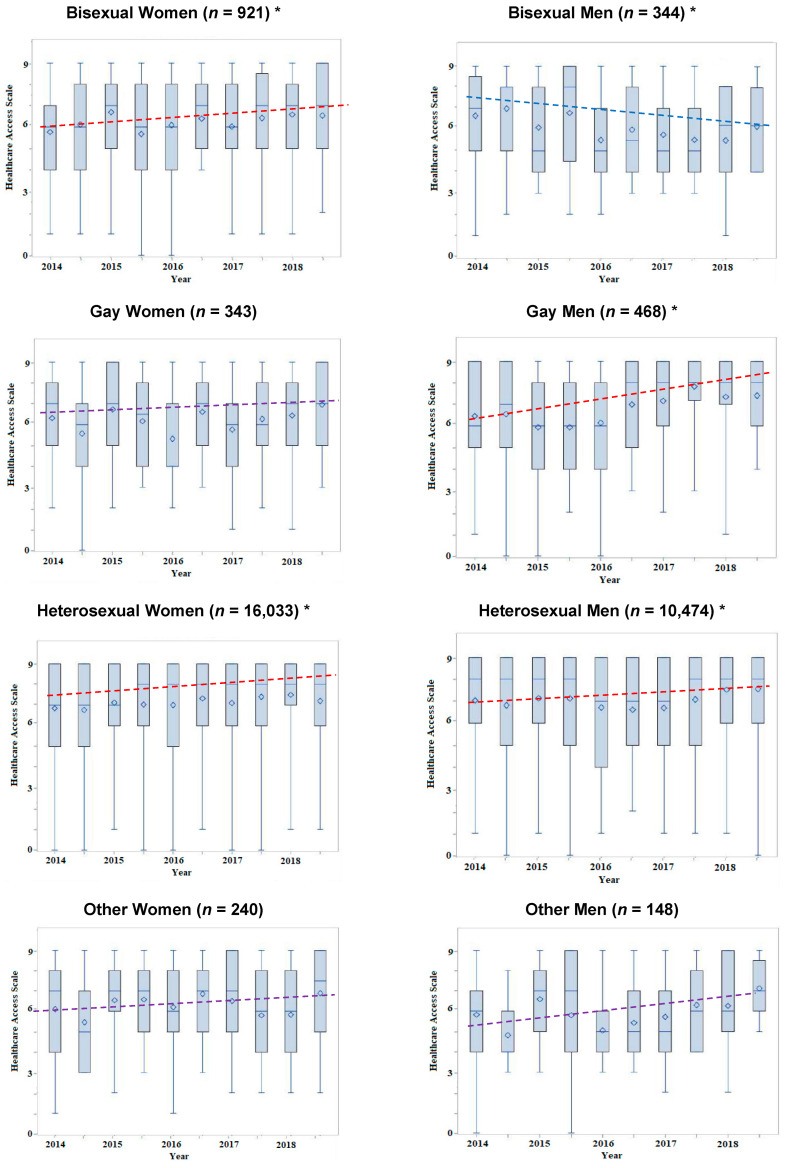
Box plots of health care access scale over time across sexual identity and sex (*n* = 28,463). * Statistically significant trend (*p* < 0.05) using Spearman rank-sum correlation. Dotted lines are linear trendline estimates. Purple lines are not statistically significant. Blue lines are sighificant decrease, and red lines are a significant increase. Boxes represent 1st quartile (bottom), median (center), mean (diamonds), and 3rd quartile (top).

**Table 1 ijerph-19-05075-t001:** Associations between time pre- and post- marriage equality (June 2015) and health care access and satisfaction measures, stratified by sexual identity (*n* = 28,961). Tested within each sexual identity using Chi-Square test. Significant results (*p* < 0.05) bolded.

	Bisexual Women (*n* = 921)		Bisexual Men (*n* = 344)		Gay Women (*n* = 343)		Gay Men (*n* = 468)		Heterosexual Women (*n* = 16,033)		Heterosexual Men (*n* = 10,474)		Other Women (*n* = 240)		Other Men (*n* = 148)	
	Pre-June 2015	June 2015 and Later	Pre-June 2015	June 2015 and Later	Pre-June 2015	June 2015 and Later	Pre-June 2015	June 2015 and Later	Pre-June 2015	June 2015 and Later	Pre-June 2015	June 2015 and Later	Pre-June 2015	June 2015 and Later	Pre-June 2015	June 2015 and Later
Current Health Insurance	83.2	86.1	91.8	95.6	88.8	87.7	85.5	82.0	88.7	90.0	89.3	91.7	**69.5**	**86.8**	90.4	88.0
Able to always get needed Health Care	**68.3**	**74.4**	78.2	81.1	76.5	77.9	**67.4**	**81.8**	83.6	87.3	85.7	88.7	70.5	66.9	66.9	67.3
Able to not Delay Health Care	65.1	61.0	**66.7**	**50.6**	61.5	57.1	**60.7**	**76.8**	75.5	75.5	68.6	65.9	52.5	61.9	51.3	48.1
Able to get Prescriptions	57.2	57.6	**61.1**	**43.9**	**66.6**	**51.8**	**61.2**	**72.6**	70.7	69.7	67.5	64.3	53.6	62.8	59.3	54.1
Able to get Medical Tests	62.1	56.1	**62.1**	**43.7**	**53.6**	**45.6**	**67.7**	**75.1**	69.3	68.6	67.6	65.1	47.5	55.0	**62.0**	**49.4**
Able to pay for needed Health Care	51.8	48.8	**52.6**	**39.4**	**53.2**	**39.2**	**55.4**	**67.3**	62.7	62.7	61.3	61.8	**42.7**	**64.6**	**70.0**	**47.3**
Satisfied with last Health Care visit	**77.3**	**84.4**	88.2	91.2	76.0	80.6	**77.4**	**86.0**	88.4	89.4	89.9	91.9	81.4	81.7	78.0	89.3
Would recommend their provider to others	69.5	73.2	85.7	85.5	71.6	73.2	77.9	77.2	81.9	84.0	83.3	86.2	76.9	74.6	67.3	77.6
Never treated unfairly due to Sexual Orientation	90.4	88.1	**80.2**	**68.5**	**88.9**	**68.3**	82.2	83.6	91.5	92.6	85.6	82.6	90.2	87.8	65.8	78.5
Never had Healthcare Barriers	18.1	19.0	**26.9**	**19.4**	20.7	19.9	**30.2**	**36.4**	34.3	36.5	35.4	34.6	14.5	19.0	**15.6**	**24.7**

**Table 2 ijerph-19-05075-t002:** Trend ratios **^1^** for time associated with health care access and satisfaction scale, stratified by sexual identity (*n* = 28,961).

	Unadjusted	Adjusted ^2^
Bisexual Women	**1.45 (1.02, 2.05)**	**1.54 (1.02, 2.32)**
Bisexual Men	**0.47 (0.22, 0.99)**	0.63 (0.32, 1.22)
Gay Women	0.81 (0.38, 1.73)	0.66 (0.32, 1.36)
Gay Men	**2.42 (1.28, 4.57)**	**1.88 (1.06, 3.34)**
Heterosexual Women	**1.17 (1.07, 1.29)**	**1.18 (1.07, 1.29)**
Heterosexual Men	**1.21 (1.08, 1.37)**	**1.23 (1.09, 1.37)**
Other Women	2.08 (0.80, 5.41)	1.17 (0.41, 3.31)
Other Men	1.26 (0.41, 3.86)	1.12 (0.34, 3.61)

Statistically significant estimates (*p* < *0*.05) bolded. Significant interaction (interaction *p* < *0*.05) between trend time and combined gender and sexual identity measure. **^1^** Trend ratios are scaled in percentage (where 0 to 100% reflects time frame from 2013 to 2018, respectively). **^2^** Models adjusted for education level, annual household income, employment status, marital status, and region.

**Table 3 ijerph-19-05075-t003:** Interrupted time series beta estimates **^1^** for trends before and after U.S. national marriage equality, associated with health care access and satisfaction scale, stratified by sexual identity and gender among sex and sexual identity (*n* = 28,961).

	Unadjusted	Adjusted ^2^
Bisexual Women	−3.17 (−6.15, 0.50)	−3.32 (−6.05, 0.71)
Bisexual Men	3.25 (−0.76, 7.25)	1.33 (−2.50, 5.17)
Gay Women	−0.07 (−4.60, 4.45)	0.64 (−3.77, 5.06)
Gay Men	**5.59 (2.00, 9.18)**	**5.06 (1.72, 8.40)**
Heterosexual Women	0.92 (0.84, 1.01)	0.88 (0.89, 1.07)
Heterosexual Men	1.09 (0.99, 1.20)	1.10 (0.97, 1.24)
Other Women	−2.78 (−8.77, 3.21)	−3.81 (−9.73, 2.10)
Other Men	0.02 (−5.52, 5.56)	−0.86 (−6.51, 4.80)

Statistically significant estimates (*p* < 0.05) bolded. **^1^** Beta estimates reflect interactions between continuous time (from 2013 to 2018, respectively) and cut-off point at June 2015. **^2^** Models adjusted for education level, annual household income, employment status, marital status, and region.

## Data Availability

This material is based upon data provided by the Association of American Medical Colleges (“AAMC”). The views expressed herein are those of the authors and do not necessarily reflect the position or policy of the AAMC. Please contact the American Association of Medical Colleges (202-828-0400) regarding data requests.

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
