# Peer review of "Trends in Health Care Access/Experiences: Differential Gains across Sexuality and Sex Intersections before and after Marriage Equality"

_ijerph, 2022, doi:10.3390/ijerph19095075_

Round 1

Reviewer 1 Report

  1. On line 37 the authors cite two studies as demonstrating higher mortality rates among for sexual minority adults.  However, the Swedish study actually showed higher mortality in Sweden only for bisexual men and women relative to heterosexuals, as there were no differences for gay men and lesbians.  Whatever the "other"category meant, those had higher mortality only for women.  
  2. The authors in that Swedish study cite (#5) Regnerus who found that the Hatzenbuehler et al. study (2014) was in error and it was eventually retracted.  Prior to retraction that study had claimed to have found a 12 year difference in mortality among sexual minorities who lived in more homophobic communities.  Since that study has been cited nearly 400 times, it is a case of how incorrect information can be spread throughout the world. 
  3. Lines 44-49 suggest that sexual minorities may be suffering health disparities due to lower incomes and/or inaffordability of healthcare.  Elwood et al.'s (2020) report in Family Relations might dispute that easy solution because in their California sample, incomes and education were substantially higher for sexual minorities, so certainly relative affordability is not an issue in all places for sexual minorities.
  4. The article seems to base its theory on sexual minority theory.  Yet there have been recent debates (see Bailey and then Meyer) about its validity or at least completeness.
  5.  Mediation theory might suggest testing for mediating factors between sexual minority status and mortality.  Research has often found that sexual minority culture in some cases at least promotes unhealthy behaviors - smoking tobacco, use of risky drugs, high rates of high BMI, etc. - that might mediate the sexual minority status variable and mortality rates.  The more challenging question would be the relative magnitude of stigma and other mediating factors in explaining the differential rates of mortality.  For the sake of sexual minorities, stigma may be uncontrollable, at least in the near term, while avoiding smoking, drug use, and overeating/underexercising might be controllable at the individual's level, regardless of larger social concerns. 
  6.  While I can see how a sexual minority person might attribute microaggressions and stigma as a function of external oppression regarding their sexual orientation, I would argue that if I had a child in school I would not tell them to avoid sexual minorities but I probably would tell them to be careful around drug users [a friend of mine's son was murdered by drug users when that son was 18 or so years old].  So if someone was gay and used drugs, my child might avoid him but it would be because of the drug use, not the sexual orientation.
  7. It's possible that physicians have issues with sexual minorities on account of health risks they take.  Heck, my own doctor gets on me about eating too much sugar, so I can see a doctor getting on a sexual minority patient on account of using drugs or being obese or whatever, not their sexual orientation per se.  But perhaps such patients would attribute the doctor's microaggressions to stigma rather than to their own risky health behaviors or conditions. 
  8.  In Table 1, it would seem that a perception of being treated unfairly on account of sexual orientation was lower or the same more recently except for one group (not always significantly).
  9.   I wonder if marital status had any role in this issue.  Elwood's study found that mental health improved for married sexual minorities but decreased for the unmarried after marriage was legalized in California.
  10. In other words, it might be useful to investigate the issues as a function of marital status by separate groups rather than simply as a control variable. 
  11. Were the trends observed for the whole sample, also the same for those of the highest socioeconomic status?  Since controlling for SES seemed to reduce the observed associations, that might indicate that higher SES was associated with better  health and possibly with sexual minority status.
  12.   Are there gender x sexual minority status x time interaction effects operating in this data set?  It seems so to me but they should be tested statistically.

Author Response

Dear Reviewer,

We are pleased to submit a second revised copy of our manuscript entitled “Trends in Health Care Access/Experiences: Differential Gains Across Sexuality and Sex Intersections Before and After Marriage Equality” to be considered for publication in the International Journal of Environmental Research and Public Health. We would like to thank all reviewers for their time spent reviewing the manuscript and their detailed feedback. We have revised the manuscript to thoroughly address each of the requests for changes. We describe our responses and revisions below each question and recommendation. Quoted comments are revisions added to the manuscript. Sections have also been included to indicate specific areas where revisions begin.

Each of the authors confirms that this manuscript has neither been published nor is simultaneously being considered for publication elsewhere. We hope that the changes we have made adequately address the reviewer’s comments and concerns about the manuscript. Thank you for considering the revised manuscript for publication in International Journal of Environmental Research and Public Health.

Reviewer 1

  1. On line 37 the authors cite two studies as demonstrating higher mortality rates among for sexual minority adults. However, the Swedish study actually showed higher mortality in Sweden only for bisexual men and women relative to heterosexuals, as there were no differences for gay men and lesbians. Whatever the "other “category meant, those had higher mortality only for women.

We have revised to specify these findings (1. Introduction):

More recent studies highlight disparities in chronic health and all-cause mortality, including greater mortality for bisexual men and women relative to their heterosexual counterparts [4, 5].”

  1. The authors in that Swedish study cite (#5) Regnerus who found that the Hatzenbuehler et al. study (2014) was in error and it was eventually retracted. Prior to retraction that study had claimed to have found a 12-year difference in mortality among sexual minorities who lived in more homophobic communities. Since that study has been cited nearly 400 times, it is a case of how incorrect information can be spread throughout the world. 

Thank you for this information.

  1. Lines 44-49 suggest that sexual minorities may be suffering health disparities due to lower incomes and/or inaffordability of healthcare.  Elwood et al.'s (2020) report in Family Relations might dispute that easy solution because in their California sample, incomes and education were substantially higher for sexual minorities, so certainly relative affordability is not an issue in all places for sexual minorities.

We have revised to note that these socioeconomic differences vary substantially across the U.S. (1. Introduction):

“Cost barriers persist despite insurance coverage gains that accompanied the passage of the Affordable Care Act and national marriage equality, though socioeconomic difficulties for sexual minorities vary substantially across regions and states, as well as across sexual identities”

Notably, California has significantly more LGBTQ+ protections and equity policy than most states, so LGBTQ+ socioeconomic status is substantially more advantaged than most of the U.S. (California consistently ranks as one of the best states for LGBTQ+ equity). Additionally, much of the key SES barriers to healthcare differ across specific sexual minority groups (such as bisexual women and bisexual men).

  1. The article seems to base its theory on sexual minority theory. Yet there have been recent debates (see Bailey and then Meyer) about its validity or at least completeness.

We have revised the discussion to note the limitations of minority stress theory, particularly when focused on sexual minorities (4. Discussion):

“Minority stress based on sexual minority identity, including discrimination and structural challenges, may not capture the full breadth of the stressors these minorities experience. Sexual minority individuals may also experience health care barriers related to race, or other identities. Future research into these intersectional identities and experiences with health care is recommended.”

  1. Mediation theory might suggest testing for mediating factors between sexual minority status and mortality. Research has often found that sexual minority culture in some cases at least promotes unhealthy behaviors - smoking tobacco, use of risky drugs, high rates of high BMI, etc. - that might mediate the sexual minority status variable and mortality rates. The more challenging question would be the relative magnitude of stigma and other mediating factors in explaining the differential rates of mortality. For the sake of sexual minorities, stigma may be uncontrollable, at least in the near term, while avoiding smoking, drug use, and overeating/under exercising might be controllable at the individual's level, regardless of larger social concerns.

We have added this as a recommendation for future research (4. Discussion):

“Future research exploring factors mediating the relationship between sexual minority identities and health care access, as well as subsequent adverse health outcomes, is recommended.”

  1. In Table 1, it would seem that a perception of being treated unfairly on account of sexual orientation was lower or the same more recently except for one group (not always significantly).

Yes, there was no group where this measure improved significantly post-marriage equality. We have revised the discussion to add this:

Marriage equality was a necessary but alone insufficient step to helping LGBTQ+ populations gain access to health insurance, but there need to be clear and targeted policies and programs to address these additional barriers highlighted in our findings, such as cost and unfair treatment due to sexual orientation; this did not significantly improve for any sexual minority groups post-marriage equality.

  1. I wonder if marital status had any role in this issue. Elwood's study found that mental health improved for married sexual minorities but decreased for the unmarried after marriage was legalized in California.

Interestingly, while we include marital status has a covariate it didn’t have a particularly large effect on the estimates, except for gay men (and even then, it was only around a ~10% difference in trend ratios, not accounting for most of the difference in unadjusted and adjusted trend estimates for this group). We have revised the manuscript to add this:

Additionally, marital status alone did not explain or account for the observed differences in healthcare access across groups.”

  1. In other words, it might be useful to investigate the issues as a function of marital status by separate groups rather than simply as a control variable. 

Similar to the above question, marital status didn’t have a significant effect on the estimates as a moderator. We had explored this earlier on but weren’t able to find much unfortunately.

  1. Were the trends observed for the whole sample, also the same for those of the highest socioeconomic status?  Since controlling for SES seemed to reduce the observed associations, that might indicate that higher SES was associated with better  health and possibly with sexual minority status.

Notably, there weren’t large differences in the overall findings across SES measures. In fact, the only group where SES was a substantial factor in the findings was for gay men (this is also evident in the reduction in trend ratios after adjusting for SES.

  1. Are there gender x sexual minority status x time interaction effects operating in this data set? It seems so to me but they should be tested statistically.

Yes, we found significant interactions between trend time and gender x sexual identity. We have added this more clearly to the footnote for Table 2:

“Significant interaction (interaction p<.05) between trend time and combined gender and sexual identity measure.”

Note that we use the cross-categorized terms for gender and sexual identity to visually highlight these differences more clearly; the interaction with time does reflect a time x gender x sexual identity interaction, however.

Reviewer 2 Report

This is a great paper. Would be nice to have more than two time points, but hey, you work with what you've got. I think the paper is sound and conforms with expectations. It did take a bit too much reading to understand whether this was a serial cross-sectional survey or a cohort so that could be mentioned more explicitly and in aims/methods/limitations as it is possible we'd find different results comparing within and between person analyses. 

Author Response

Reviewer 2

This is a great paper. Would be nice to have more than two time points, but hey, you work with what you've got. I think the paper is sound and conforms with expectations. It did take a bit too much reading to understand whether this was a serial cross-sectional survey or a cohort so that could be mentioned more explicitly and in aims/methods/limitations as it is possible we'd find different results comparing within and between person analyses. 

Thank you for the interest in the paper! We appreciate the recommendations, and have added that this is a serial cross-sectional study more explicitly in both the methods (2.1 Sample):

“This is a serial cross-sectional survey of US national sample of respondents who reported needing health care in the last 12 months.”

…and to the limitations paragraph (4. Discussion):

“First, this was a serial cross-sectional study; future research using prospective data is needed to assess within-person differences in health care access and satisfaction over time.”

Reviewer 3 Report

In this manuscript, the authors describe use of large population-based samples collected biannually from 2013 to 2018 (11 waves) to explore trends in health care access and satisfaction, and differences in trends by sex and sexual identity. They also explore whether the passage of federal marriage equality in 2015 correspond to changes in health care access and satisfaction. This is a timely and important topic, with potential to inform our understanding of how policies can influence health care access among sexual minorities, an important source of the health disparities they face. Strengths include the large and representative sample and examination of different demographic groups within sexual minorities, rather than lumping them all together as many studies do, which would have missed key differences in their experiences. I have some concerns about the manuscript in its current form that weaken its potential contribution to the literature, most related to a lack of specificity in how variables were coded and analyses were run, which prohibits me from fully evaluating whether they were done appropriately and interferes with interpretation of findings. I list my specific concerns below:  

  1. The title is a little misleading, as it suggests that all of the data is post- marriage equality.
  2. I wonder if it is appropriate to use both the access and satisfaction items within one scale. Though the reasonably high alpha for the combined scale does argue for this, it would also be possible to get that alpha even if there were really two different constructs being tapped by the scale. The authors should assess for unidimensionality of the scale to support this approach or consider using the access and satisfaction items separately.
  3. More details are needed about the data and analyses for the reader to be able to evaluate their rigor and interpret the results:
    1. The authors should clearly state how many time points were considered pre- and post- marriage equality. Also, the dates of data collection are not consistent across the paper—in different places the start date is 2012, 2013, or 2014.
    2. Though the Ns for each gender/sexual identity group are fairly large, these Ns appear to be an aggregate across all 11 waves. Since data analysis includes attention to health care access and satisfaction at each time point, information is needed about the Ns by timepoint. If only 8.7% of each wave is non-heterosexual, and each wave had as few as 2,000 participants, I wonder about the number of people in each group at each time point- are they large enough to produce stable estimates?
    3. It is unclear what the values presented in Table 1 are. How were the multiple time points before and after June 2015 combined into one dichotomous variable for each item? The sentence describing these analyses: “For bivariate analyses, we tested if time post-marriage equality was associated with each healthcare/satisfaction measure using Chi-Square tests” is not sufficient since it is not clear how either time or access/satisfaction were operationalized.
    4. Then, was time coded differently for these analyses: “We also tested for associations between time and our healthcare access/satisfaction scale using a Spearman rank-sum correlation test.”?
    5. In the rest of the analytic plan, numerous analyses are described—but it is not clear how these map onto the research questions, or which results come from which analyses. This needs to be made much more clear.
    6. It would be useful to assess for differences between specific groups in the time slopes for access/satisfaction presented in Table 1. In particular, were changes in access/satisfaction greater for sexual minorities than heterosexuals? This would help address issues raised in the discussion that access/satisfaction may have improved for all Americans, perhaps due to the Affordable Care Act.
    7. It is unclear how to interpret the coefficients presented in Table 3. A better description of the analysis run will help, but also the authors need to guide the reader to understand what the time trends are before and after the cut-point for each group, and if there is evidence that the slope changed at that cut-point.
  4. From what I can tell, the only time effects tested were linear. Did the authors evaluate whether a linear model was most appropriate, over other models?

Author Response

Dear Reviewer,

We are pleased to submit a second revised copy of our manuscript entitled “Trends in Health Care Access/Experiences: Differential Gains Across Sexuality and Sex Intersections Before and After Marriage Equality” to be considered for publication in the International Journal of Environmental Research and Public Health. We would like to thank all reviewers for their time spent reviewing the manuscript and their detailed feedback. We have revised the manuscript to thoroughly address each of the recommended changes. We describe our responses and revisions below each question and recommendation. Quoted comments are revisions added to the manuscript. Sections have also been included in parentheses to indicate specific areas where revisions begin.

Each of the authors confirms that this manuscript has neither been published nor is simultaneously being considered for publication elsewhere. We hope that the changes we have made adequately address the reviewer’s comments and concerns about the manuscript. Thank you for considering the revised manuscript for publication in International Journal of Environmental Research and Public Health.

Reviewer 3

In this manuscript, the authors describe use of large population-based samples collected biannually from 2013 to 2018 (11 waves) to explore trends in health care access and satisfaction, and differences in trends by sex and sexual identity. They also explore whether the passage of federal marriage equality in 2015 correspond to changes in health care access and satisfaction. This is a timely and important topic, with potential to inform our understanding of how policies can influence health care access among sexual minorities, an important source of the health disparities they face. Strengths include the large and representative sample and examination of different demographic groups within sexual minorities, rather than lumping them all together as many studies do, which would have missed key differences in their experiences. I have some concerns about the manuscript in its current form that weaken its potential contribution to the literature, most related to a lack of specificity in how variables were coded and analyses were run, which prohibits me from fully evaluating whether they were done appropriately and interferes with interpretation of findings. I list my specific concerns below.

Thank you for the interest in the paper! We appreciate the recommendations and have described our point-by-point revisions based on your recommendations below.

  1. The title is a little misleading, as it suggests that all of the data is post- marriage equality.

We have changed the title to “Trends in Health Care Access/Experiences: Differential Gains Across Sexuality and Sex Intersections Before and After Marriage Equality” to clarify this.

  1. I wonder if it is appropriate to use both the access and satisfaction items within one scale. Though the reasonably high alpha for the combined scale does argue for this, it would also be possible to get that alpha even if there were really two different constructs being tapped by the scale. The authors should assess for unidimensionality of the scale to support this approach or consider using the access and satisfaction items separately.

Thank you for this recommendation; we have added a revision to clarify that we have conducted a factor analysis, with findings consistent with unidimensionality (2.3 Health care access and satisfaction):

“Additionally, factor analysis demonstrated a single factor for all items, consistent with unidimensionality.”

  1. More details are needed about the data and analyses for the reader to be able to evaluate their rigor and interpret the results. The authors should clearly state how many time points were considered pre- and post- marriage equality.

We have added this to the methods (2.2 Sexual identity, sex, and time): For initial bivariate measures, we used a dichotomized measure of time, stratified in June 2015. The first 5 waves were included as pre-marriage equality (before June 2015), and last 6 were included as post-marriage equality (June 2015 and after). For correlation and regression analyses, we used the uncategorized measure from the 11 waves, with each wave collected 6 months apart.”

  1. Also, the dates of data collection are not consistent across the paper—in different places the start date is 2012, 2013, or 2014.

We have corrected the sample description (2.1 Sample) and Tables 2 and 3 to clarify that the start date for the data is 2013.

  1. Though the Ns for each gender/sexual identity group are fairly large; these Ns appear to be an aggregate across all 11 waves. Since data analysis includes attention to health care access and satisfaction at each time point, information is needed about the Ns by timepoint. If only 8.7% of each wave is non-heterosexual, and each wave had as few as 2,000 participants, I wonder about the number of people in each group at each time point- are they large enough to produce stable estimates?

Thankfully, the use of cumulative ordinal modeling is able to maintain stability even with a relatively small cell size per time stratum. Even with just around 20 individuals per timepoint we can generate very stable estimates. This was achieved for all sex and sexual identity groups with one exception: the “other men” category. Though we were still able to achieve convergence for this group, the standard errors were quite large, and there are significant limitations to inferences; we have added this as a limitation (Discussion):

“Lastly, the study was limited in the ambiguity that the “other” sexual orientation category introduced. Given the small number of individuals in this group (particularly the “other men” group) it is difficult to make inferences about this population.”

  1. It is unclear what the values presented in Table 1 are. How were the multiple time points before and after June 2015 combined into one dichotomous variable for each item? The sentence describing these analyses: “For bivariate analyses, we tested if time post-marriage equality was associated with each healthcare/satisfaction measure using Chi-Square tests” is not sufficient since it is not clear how either time or access/satisfaction were operationalized.

We’ve revised to clarify how time was coded here (2.5 Analyses):

“For these analyses only, time was coded as a binary measure, with all time values grouped into one of two categories: Any time before June 2015, or June 2015 and after.”

Note that we only use dichotomous time for this table, to present prevalence of healthcare access and satisfaction before and after marriage equality.

  1. Then, was time coded differently for these analyses: “We also tested for associations between time and our healthcare access/satisfaction scale using a Spearman rank-sum correlation test.”?

Yes, we used discrete time (each wave uncollapsed) for these analyses. We have revised to clarify (2.5 Analyses):

“We also tested for associations between the original discrete time measure and our healthcare access/satisfaction scale using a Spearman rank-sum correlation test.”

We also discuss this earlier in the methods (2.2 Sexual identity, sex, and time):

“For correlation and regression analyses, we used the uncategorized measure from waves (ranging from 7 to 18, with each wave collected 6 months apart).”

  1. In the rest of the analytic plan, numerous analyses are described—but it is not clear how these map onto the research questions, or which results come from which analyses. This needs to be made much clearer.

We have revised to clarify the purpose for these tests:

“Finally, to visualize trends in healthcare access and satisfaction, we generated box plots with ordinal trendlines for our healthcare access/satisfaction scale. All bivariate analyses were stratified by sex and sexual identity to examine differences in healthcare access and satisfaction across sex and sexual identity.”

“To determine if healthcare access/satisfaction trends were significantly different across sex and sexual minority groups, we generated a single model with interaction terms testing sex and sexual identity as an effect modifier of the trend in healthcare access.”

“To examine if trends in healthcare access and satisfaction changed for sexual minority groups after marriage equality, we also used an interrupted time-series regression model to test for a difference in in trends of healthcare access before and after the post-marriage equality cut point (June 2015).”

  1. It would be useful to assess for differences between specific groups in the time slopes for access/satisfaction presented in Table 1. In particular, were changes in access/satisfaction greater for sexual minorities than heterosexuals? This would help address issues raised in the discussion that access/satisfaction may have improved for all Americans, perhaps due to the Affordable Care Act.

We have added both unadjusted and adjusted trend ratios for heterosexual women and men to Table 3. Notable, none were significant, as they were all very close to 1.00, suggesting no substantial change in healthcare access and marriage equality for heterosexuals. We’ve added this to the results as well:

Notably, gay men had a significant increase in health care access trend after U.S. national marriage equality based on interrupted time series estimates (unadjusted = 5.59, 95% CI 2.00, 9.18; adjusted = 5.06, 95% CI 1.72, 8.40), while other sexual minority groups and heterosexuals did not (Table 3).”

  1. It is unclear how to interpret the coefficients presented in Table 3. A better description of the analysis run will help, but also the authors need to guide the reader to understand what the time trends are before and after the cut-point for each group, and if there is evidence that the slope changed at that cut-point.

We have added a clearer description of the time trends, with an example (3.2 Regression and Interactions):

Here, interrupted time-series estimates reflect the direction of the change in trends, and confidence intervals that exclude 1 indicate statistical significance. The positive estimate for gay men (5.06) indicates an increase in the trend in healthcare access, with statistical evidence (p<.05) that the slope trend in health care access became more positive after the marriage equality cut-off (June 2015).”

In general, interpreting the time trends based on positivity/negativity,

  1. From what I can tell, the only time effects tested were linear. Did the authors evaluate whether a linear model was most appropriate, over other models?

Yes, we assessed validity of linear terms for time by checking for significant deviations from linearity. First, we examined estimates for healthcare access and satisfaction across each individual timepoint. We also compared estimates to a theoretical estimate distribution with perfect linearity, checking for patterns in residuals between our observed estimates and the theoretical model. Overall, there was no substantial difference in estimates (<10%) between the observed and theoretically linearized model, supporting the use of linear time effects.

Round 2

Reviewer 1 Report

The authors did a great job of improving the manuscript, thanks to them for such a good revision!

Author Response

Thank you so much!

Reviewer 3 Report

The authors did a good job addressing the concerns I raised about the original version of the manuscript. The additional information about the data, analyses, and how to interpret Table 3 has significantly improved the paper. I have only two minor remaining comments:

  1. In lines 184-187, the description of results regarding the item, "never had health care barriers" is a bit confusing. It reads as though other sexual minority men had increased in healthcare barriers, but I think the data suggest that the proportion who NEVER had barriers increased from 15.6 to 24.7%, indicating the effect is in the opposite direction. Same comment for bisexual men. Please clarify these findings.
  2. In lines 226-7 the authors state that bisexual women had a decrease in healthcare access trends post-marriage equality. I couldnt find this finding in the results section- is it from a previous version of the paper and needs to be removed? Or else, please clearly describe the finding supporting this comment. 

Author Response

Dear Reviewer,

We are pleased to submit a third revised copy of our manuscript entitled “Trends in Health Care Access/Experiences: Differential Gains Across Sexuality and Sex Intersections Before and After Marriage Equality” to be considered for publication in the International Journal of Environmental Research and Public Health. We have revised the manuscript to thoroughly address the two recommended changes, describing our responses and revisions below each question and recommendation. Quoted comments are revisions added to the manuscript. Sections have also been included in parentheses to indicate specific areas where revisions begin.

Each of the authors confirms that this manuscript has neither been published nor is simultaneously being considered for publication elsewhere. We hope that the changes we have made adequately address the reviewer’s comments and concerns about the manuscript. Thank you for considering the revised manuscript for publication in the International Journal of Environmental Research and Public Health.

  1. In lines 184-187, the description of results regarding the item, "never had health care barriers" is a bit confusing. It reads as though other sexual minority men had increased in healthcare barriers, but I think the data suggest that the proportion who NEVER had barriers increased from 15.6 to 24.7%, indicating the effect is in the opposite direction. Same comment for bisexual men. Please clarify these findings.

We thank you for catching this error, and have corrected it as follows (3.1 Sample Characteristics and Bivariate Results):

“The largest increase in proportions of having no healthcare barriers was observed among other sexual minority men (15.6% to 24.7%). In contrast, bisexual men had significant decreases in health care access across almost every single item and decreased proportions of having no healthcare barriers (26.9% to 19.4%).”

  1. In lines 226-7 the authors state that bisexual women had a decrease in healthcare access trends post-marriage equality. I couldn’t find this finding in the results section- is it from a previous version of the paper and needs to be removed? Or else, please clearly describe the finding supporting this comment.

We thank you for catching this error and have deleted this erroneous sentence.
